# Challenge in the Discovery of New Drugs: Antimicrobial Peptides against WHO-List of Critical and High-Priority Bacteria

**DOI:** 10.3390/pharmaceutics13060773

**Published:** 2021-05-21

**Authors:** Cesar Augusto Roque-Borda, Patricia Bento da Silva, Mosar Corrêa Rodrigues, Ricardo Bentes Azevedo, Leonardo Di Filippo, Jonatas L. Duarte, Marlus Chorilli, Eduardo Festozo Vicente, Fernando Rogério Pavan

**Affiliations:** 1Tuberculosis Research Laboratory, School of Pharmaceutical Sciences, São Paulo State University (UNESP), Araraquara 14800-903, Brazil; cesar.roque@unesp.br; 2Laboratory of Nanobiotechnology, Department of Genetics and Morphology, Institute of Biological Sciences, University of Brasilia, Brasilia 70910-900, Brazil; patrbent@gmail.com (P.B.d.S.); mosarcr@gmail.com (M.C.R.); razevedo@unb.br (R.B.A.); 3Department of Drugs and Medicines, School of Pharmaceutical Sciences, São Paulo State University (UNESP), Araraquara 14800-903, Brazil; leonardo.filippo@unesp.br (L.D.F.); jl.duarte@unesp.br (J.L.D.); marlus.chorilli@unesp.br (M.C.); 4School of Sciences and Engineering, São Paulo State University (UNESP), Tupã 17602-496, Brazil; eduardo.vicente@unesp.br

**Keywords:** AMPs, antibacterial activity, critical-priority bacteria, high-priority bacteria, MDR, XDR

## Abstract

Bacterial resistance has intensified in recent years due to the uncontrolled use of conventional drugs, and new bacterial strains with multiple resistance have been reported. This problem may be solved by using antimicrobial peptides (AMPs), which fulfill their bactericidal activity without developing much bacterial resistance. The rapid interaction between AMPs and the bacterial cell membrane means that the bacteria cannot easily develop resistance mechanisms. In addition, various drugs for clinical use have lost their effect as a conventional treatment; however, the synergistic effect of AMPs with these drugs would help to reactivate and enhance antimicrobial activity. Their efficiency against multi-resistant and extensively resistant bacteria has positioned them as promising molecules to replace or improve conventional drugs. In this review, we examined the importance of antimicrobial peptides and their successful activity against critical and high-priority bacteria published in the WHO list.

## 1. Introduction

In recent decades, uncontrolled drug use has led to major public health, economic, and environmental problems. As a result, bacteria, as well as other microorganisms, have acquired or developed unfavorable resistance mechanisms toward conventional drugs [1]. The World Health Organization (WHO) has launched a priority list of bacteria that should be studied more rigorously, and all the bacteria on this list are resistant to various drugs [2]. Bacterial resistance is a global public health concern, and it becomes a serious problem when the line of resistance exceeds the inactivation of first-line drugs, which consequently leads to high mortality rates [3]. This problem arises when a bacterium begins to generate defense molecules or acquires resistance plasmids from other bacteria, which is why, in recent years, the use of drugs has been urgently restricted [4]. For this reason, in the search for new drugs, antimicrobial peptides (AMPs) have been highlighted.

AMPs are biomolecules produced as a defense mechanism of living beings, with excellent antibacterial, antifungal, and antiviral properties [5]. They are distinguished by their ability to greatly reduce the bacterial load to low concentrations and to generate a minimal response of bacterial resistance due to the rapid action with lipopolysaccharides (LPS) of the cell membrane [6]. This article reviews the growing number of studies that demonstrate the efficacy of antimicrobial peptides against major groups of dangerous bacteria, as potential new substituents for conventional drugs (Figure 1).

## 2. Why Are AMPs Highlighted in the Development of New Antibiotics?

Innate immunity is the oldest defense system against invading microorganisms. It is the first-line mechanism to virtually protect all multicellular organisms from pathogens. Particularly, a set of biomolecules from the mammalian innate immune system is notably similar to components in other kingdoms, such as plants and insects. In this way, evidence indicates that these molecules—AMPs—could have emerged in common ancestors a long time ago, in the evolutionary processes. Defensins, a classical example of very active AMPs against fungi and bacteria, are found in plants and mammals and have essentially the same tridimensional structure in both organisms [7]. AMPs present a broad structural and functional diversity. Over the last decade, researchers have shed light on AMPs, notably receiving attention for the development of new and natural antibiotic candidates (Figure 2). AMPs are interesting for treating infections, mainly those caused by the so-called and dangerous multidrug-resistant (MDR) bacteria. Nowadays, infections caused by this group of bacteria have reached a pan-drug resistance level, which means resistance to all available conventional antibiotics. Curiously, this scenario was predicted by the penicillin discoverer, Alexander Fleming, who received a Nobel Prize in 1945 and anticipated that the global misuse of antibiotics in several areas could generate a frightening panorama [8].

AMPs share typical features, such as 8 to 50 amino acids, in the primary sequence. A cationic net charge along the structure is often observed, which can induce and facilitate an initial interaction with membrane headgroup components. There is also a predominance of hydrophobic residues, up to 50% of the entire sequence, allowing van der Waals interactions with the membrane lipid tails [9,10]. AMPs can adopt a broad variety of secondary structures: the helical magainin; the protegrin, which forms a β-hairpin due to the presence of a disulfide bond; the disordered indolicidin; and gramicidin, which can be cyclic [11].

AMPs display several mechanisms of action to combat pathogens. These mechanisms depend on physicochemical features such as primary and secondary structures, the number of residues, net charge, and amphipathicity [12]. Overall, they can be classified according to these properties into membrane lytic and nonlytic AMPs. There are four well-established models of membrane lytic peptides in the literature: (i) the carpet model, whereby AMPs bind to an anionic target membrane like a “carpet,” covering the surface. Peptides strongly interact with the lipid head groups throughout the membrane, in a detergent-like disruption process; (ii) the toroidal pore or wormhole model, whereby the AMPs first induce the inner and outer membrane leaflet curvature and, as a consequence, align the pore lumen parallel to the phospholipid orientation, along with the headgroups. A variation of this model is the disordered toroidal pore, performed by magainin analogs and melittin, the pore formation being more stochastic and involving fewer peptide monomers; (iii) the barrel-stave pore or helical bundle model, in which the peptide monomers insert into the membrane, forming a barrel shape and open stables and transmembrane pores, which destabilize the membrane potential and promote ion and biomolecule leakage; (iv) the aggregate channel model comprises a competitive replacement of LPS-associated divalent cations, causing an unstructured aggregation of peptides and lipids, which disrupt the outer and inner membranes. In addition, other lytic membrane models, such as the lipid segregation model, oxidized lipid targeting, changes in membrane potential, and electroporation have been unveiled and described. Regarding the nonmembrane lytic peptide mechanism of action, AMPs mostly target intracellular structures, binding to DNA gyrase and topoisomerase IV, directly affecting transcription/replication. In addition, AMPs can inhibit protein and cell wall biosynthesis, inactivate enzyme activities in ribosomes, and cause cellular apoptosis [9,11,13].

AMPs can be obtained by chemical synthesis or recombinant production systems. Specifically, the latter typically demands a long and expensive research and development phase, exhibiting some limitations in terms of modifications in the peptide sequence. However, the artificial synthesis of AMPs provides interesting advantages from natural or recombinant obtainment processes. In the solid-phase peptide synthesis (SPPS), amino acids are sequentially added to a solid support, which is considered the most commonly used and mature technology available for AMPs, even for the production of peptides with up to 50 amino acid residues [14]. A detailed approach to the SPPS method is described elsewhere [15]. Interestingly, SPPS offers a precise modification of AMP primary sequences, to modulate their biological activity against different types of pathogens and evaluate structure–activity relationships. Moreover, the design of AMPs via chemical synthesis can incorporate unusual and nonnatural amino acids to study the backbone conformation, dynamics in the membrane, and solution and orientation of peptides [16,17], via spectroscopy techniques [18,19].

Although AMPs are considered a valuable candidate for a new line of antibiotic compounds to tackle bacterial resistance, there are still many challenges to be overcome: (i) poor biostability by proteolytic degradation; (ii) cytotoxicity/high hemolytic activities at concentrations closer to therapeutic dosages; (iii) the lack of efficient delivery systems to the target site for effective release concentrations; and (iv) high manufacturing costs. In this way, the main objective of researchers and biotechnological and pharmaceutical companies is the production of AMPs that are highly selective against pathogenic bacteria, exhibiting higher therapeutic indexes. There are currently several successful AMPs in phase 3 of clinical trials, such as Omiganan, IMX942 (topical application); Surotomycin and Talactoferrin (oral); and Murepavadin and p2TA (intravenous) [11,20]. These AMPs have different targets and mechanisms of action and have aroused high expectations about the development and registration of safer, efficient, and natural antibiotics to combat MDR bacteria and avoid a chaotic future scenario.

## 3. Critical Bacterial Resistance

The great global concern regarding bacterial resistance to drugs has led the WHO to publish priority lists of bacteria for research and new drugs that may help reduce or control this problem. This list has been separated into critical-priority (CPB), high-priority (HPB), and medium-priority bacteria [21] according to the resistance of the antimicrobials currently used.

In its latest updates, between 2015 and 2017, the FDA approved the use of avibactam, bezlotoxumab, ceftazidime, delafloxacin, malacidin, obiltoxaximab, ozenoxacin, teixobactin, vabomere, and vaborbactam [22], and in 2018–2019, cefiderocol, cilastatin, relebactam, eravacycline, imipenem, lefamulin, omadacycline, rifamycin, plazomicin, pretomanid, sarecycline [23]. The mechanisms of action and antimicrobial activity of the new drugs are described in Table 1, published by the FDA between 2017 and 2021.

For a better understanding of the bacterial resistance of this priority list of bacteria, the classification of conventional drugs used worldwide is shown in Table 2 (where the new drugs mentioned above are not listed).

The bacterial resistance to beta-lactam and quinolone drugs is highlighted on the CPB list, leading to strong interest in developing new drugs and pharmaceutical combinations [26]. Beta-Lactam drugs are a broad class of antibiotics consisting of a β-lactam ring and a broad spectrum, which act as inhibitors of cell wall formation, preventing peptidoglycan biosynthesis [27]. All antibiotics of this class (except for tabtoxinine-β-lactam, which inhibits glutamine synthetase) bind to penicillin-binding proteins (PBPs), a type of transpeptidase responsible for the synthesis of the bacterial cell wall [28]. Resistance to carbapenems is the main concern of the WHO [26]. Carbapenem drugs (β-lactam group) are used in the treatment of highly uncontrollable multidrug-resistant (MDR) bacterial diseases. Their action also allows the blocking of cell wall formation by interrupting peptidoglycan transpeptidation in the cell wall [29]. Over the years, several drugs of this subgroup have been discovered and approved, such as imipenem (1985), meropenem (1993), panipenem/betamipron (1993), ertapenem (2001), doripenem (2007), biapenem (2001), and tebipenem (2015). Despite being excellent bacterial inhibitors, many bacterial strains have developed resistance [30].

The first report of bacterial resistance to β-lactams was on penicillins (1940) through hydrolysis reaction, efflux pumps, and alterations of the target, producing, in many cases, certain enzymes known as β-lactamases, which inactivate the action of these drugs (hydrolyzing β-lactam bonds) [27]. Other studies prioritize AMP-drug activity (synergic effect) to enhance the action of carbapenems, as recently reported, using meropenem-vaborbactam to kill high-risk bacteria with resistance to carbapenems. Vaborbactam is a boron-based inhibitor that inactivates the action of *Klebsiella pneumoniae* carbapenemase (KPC), which does not have antibacterial activity when used unimolecularly [31], Imipenem-cilastatin [32], and Imipenem-cilastatin-relebactam [32,33]. The combined activity or synergistic effect has had positive results in recent years, and its mechanisms still need to be studied in depth.

Resistance to fluoroquinolones was also highlighted out on this list because different drugs, such as ciprofloxacin, were used to treat infectious diseases such as *Moraxella catarrhalis*, *Haemophilus influenzae*, *Streptococcus pneumoniae*, *Neisseria gonorrhoeae*, and *Enterobacteria*. Due to their uncontrolled use and/or the discontinuation of treatment, new mutations were reported worldwide [34,35,36]. The action of fluoroquinolones occurs by the accumulation of -OH radicals, chromosomal fragmentation, and an increase in reactive oxygen species. However, similar to carbapenems, quinolone-resistant bacteria also present resistance mechanisms such as efflux pumps and enzymes, diverting the drug target, reducing drug accumulation, and/or generating defense plasmids [37]. AMP-drug activity was also reported to have a synergistic effect between Melimine, Protamine, and Mel4, together with ciprofloxacin and cefepime, which resulted in the elimination of all MDR bacteria with lower minimal inhibition concentration (MIC) values, thus enhancing the use of these two drugs, which, by themselves, lose their action against MDR bacteria [38].

Other drug resistances shown in the WHO list involve glycopeptides, which are frequently used against enterococci, streptococci, and MDR staphylococci. Among them, vancomycin and teicoplanin, derived from Actinobacteria, are included on the HPB list [39]. These drugs act by binding to the peptidoglycan ends of the cell membrane (binding between the d-Ala-d-Ala dipeptide and Lipid II precursor); this reduces the activity of transglycosylases and transpeptidases, thus blocking bacterial replication caused by a metabolic imbalance, until cell death [40]. Telavancin, Dalbavancin, and Oritavancin were the last drugs approved and FDA-incorporated as treatments, as reported until 2018 [39]. However, bacterial resistance in this group of drugs occurs mainly to vancomycins, when these bacteria replace the production of d-Ala-d-Ala by d-Ala-X, X being a d-Lac (vanA, vanB, and vanC) or d-Ser, which removes one of the five bonds formed by vancomycin, and, therefore, they lose their affinity and effectiveness [39]. This effect improved when a conjugate of vancomycin with cationic AMP, known as FU002, was used. This conjugate made it possible to drastically decrease the MIC values in several *Staphylococcus aureus* methicillin-resistant (MRSA) strains, increasing its effect 1000 times and achieving a profile of controlled biodistribution [41]. For an in-depth review of glycopeptide-resistant bacteria, see Blaskovich et al. [39].

Likewise, AMPs have shown strong performance against bacteria resistant to macrolides such as clarithromycin. This resistance is generally caused by mutations alleged to *Helicobacter pylori*, which is installed in the gastric mucosa and is the main cause of stomach cancer in the world [42]. This macrolide binds to the 50S subunit, inhibiting protein synthesis and, consequently, its replication. Resistance to macrolides is produced by the genetic alteration of rRNA 23S and the production of efflux pumps, which hinders the action mainly of clarithromycin [43]. AMP T4 was described as a promising peptide because it acts independently of the type of resistance, evaluated against *H. pylori* MDR and showing better results than the drugs usually used [44].

In summary, the application and design of new AMP or AMP homologs are necessary to meet the demand for new drugs against resistant bacteria. Various studies of AMPs against these MDR, extensively drug-resistant (XDR), and pan-drug resistant (PDR) bacteria were reviewed and classified according to their location on the WHO-list (CPB and HPB).

## 4. AMP Applications in Critical-Priority Bacteria

The CPB list groups *Acinetobacter baumannii* and *Pseudomonas aeruginosa*, carbapenem-resistants, Enterobacteriaceae (*Klebsiella pneumoniae*, *Escherichia coli*, *Enterobacter* spp., *Serratia* spp., *Proteus* spp., *Providencia* spp., and *Morganella* spp.), carbapenem-resistant (CRE), third-generation cephalosporin-resistant, and Mycobacterium tuberculosis [21].

All the bacteria on this list show severe antimicrobial resistance, and most reported cases are found in patients admitted to intensive care units (ICUs). *Acinetobacteria* sp. are a special group of microorganisms present in the ICU that easily acquire resistance to antibiotics during the treatment of patients hospitalized for other diseases. In particular, *A. baumannii* causes pneumonia and dermal and urethral infections [45]. This large-negative bacterium, in the form of a coccobacillus, belongs to the group of opportunistic bacteria and the reservoir of resistance genes that is more difficult to control [46]. Due to these parameters, carbapenems were the most efficient treatment against these bacteria until the last decade, when new bacterial strains with resistance to carbapenems were reported, which was even more worrying [29].

*A. baumannii* can generate resistance by enzymatic or nonenzymatic routes. It was observed in recent years that the alteration of PBPs would be involved in a mechanism of resistance to β-lactam drugs and, consequently, to carbapenems [47,48]. In addition, the ratio of the decrease in the size of the outer membrane proteins (OMP), known in *A. baumannii* as CarO proteins, would be the main way to identify their nonenzymatic resistance. Another means would be the presence of active expulsion or efflux pumps, which are the main source of bacterial resistance in all MDR bacteria [29]. Commonly used drugs for this type of MDR bacteria are colistin sulfate, tobramycin, levofloxacin, kanamycin, and carbenicillin disodium [49]. Different successful studies were carried out to eliminate this bacterium with AMPs, as listed in Table 2.

*P. aeruginosa* causes lung inflammation due to the release of IL-6 and TNF-α induced by their LPS, which are associated with cystic fibrosis and chronic obstructive pulmonary disease [49]. They are difficult bacteria to treat, and like *A. baumannii*, many of those with MDR or XDR are biofilm-forming, which prevents the action of most drugs [50]. In the same way, combined studies of AMP-drugs have been carried out to improve their bactericidal activity, and even AMPs have been considered new biomolecules with antibiofilm action, which allows the promotion of the development of new drugs for this list of potentially infectious bacteria [51,52].

*Enterobacteriaceae* have become a family of problematic bacteria due to the ease of resistance and the production of extended-spectrum β-lactamases and carbapenemases (KPC and others), which induce carbapenem-resistant Enterobacteriaceae (CRE) infections [53,54]. *Klebsiella pneumoniae* MDR is a highly complex bacterium to treat and, due to its persistence, the mortality rate in humans continues to increase. The use of polymyxin (polypeptides approved by the FDA) has been intensified to combat many MDRs, as this peptide helps to improve the action of obsolete drugs during treatment [54]. A study revealed that there are certain amino acid conformations of peptides that could efficiently fight this type of MDR/XDR/PDR bacteria by modifying the conformation of this capsule, although there is an unknown information barrier to explain this event that must be explored [55]. An in-depth study of CRE was reported by Suay-García et al. [54].

*Mycobacterium tuberculosis* is a bacterium that causes a high mortality rate worldwide. It has distinct defense mechanisms, in comparison with other bacteria, due to its capsule composition of peptidoglycans. It is also capable of remaining in the latency phase for many years with no symptoms of infection [56]. Thus, many studies against this bacterium have been carried out at the genetic-molecular and biochemical levels to reach new therapeutic objectives and control it, as conventional antibiotics are not effective enough, require a long time, and cause unpleasant adverse effects [57]. A study reported that the regulation between the relationship of cortisol and dehydroepiandrosterone in the human body could be related to the human immune response to tuberculosis, and this relationship could increase or decrease the production of innate AMPs, such as cathelicidin LL-37, and human β-defensin 2 and 3, to combat this bacterium [58]. The use of AMPs against tuberculosis or the regulation of their innate production is an attractive option that has created great expectations.

In a recent study, Abraham et al. [56] synthesized a 21-residue peptide called B1CTcu5, which was originally isolated from the cutaneous secretion of *Clinotarsus curtipes*, a frog of Indian origin. The brevinin-1 family of AMP has shown promising results in the search for a new agent against tuberculosis. It was able to selectively kill intracellular bacteria without causing damage to macrophages and had an overall hydrophobicity of 65%. The hydrophobic nature of the peptide may act as a driving force to move from an aqueous environment to a hydrophobic one and increase its affinity for the acyl lipid chains of the membrane. The hydrophobic interaction between amphipathic AMP and the cell membrane forms a specific peptide–lipid complex, which can produce changes in the bacterial membrane, such as thinning, pore formation, altered curvature, and localized alterations. The AMP can translocate across the membrane and diffuse into the cytoplasm, possibly interacting with intracellular targets. According to Orme et al. [59], potential compounds should have <10 µg/mL MIC values. Most therapeutic drugs are effective below 1 µg/mL. Table 3 shows AMPs designed to have great antimicrobial activity against CPB. The analysis was based on a systematic review in the ScienceDirect database and Google scholar search engine in 2019–2021.

## 5. AMP Applications against High-Priority Bacteria

This group includes the following bacteria: *Enterococcus faecium*, vancomycin-resistant; *Staphylococcus aureus*, methicillin-resistant, vancomycin intermediate, and resistant; *Helicobacter pylori*, clarithromycin-resistant; *Campylobacter*, fluoroquinolone-resistant; *Salmonella* spp., fluoroquinolone-resistant; *Neisseria gonorrhoeae*, third-generation cephalosporin-resistant, fluoroquinolone-resistant.

### 5.1. Vancomycin-Resistant Enterococcus Faecium (VRE)

Anoplin (GLLKRIKTLL-NH_2_) is a peptide isolated from the venom of the Japanese solitary spider wasp *Anoplius samariensis*. A study conducted by Munky et al. [101] evaluated the antimicrobial activity of Anoplin and its analogs. The AMP analogs were synthesized by substituting at 2, 3, 5, 6, 8, 9, and 10 amino acid positions. The synthesized analogs were peptides modified at position 5 (Lys^5^ and Trp^5^), position 8 (Lys^8^), positions 5 and 8 together (Lys^5^Lys^8^, Lys^5^Trp^8^, Trp^5^Lys^8^, Phe^5^Lys^8^, and Phe^5^Trp^8^), β-2-naphthylalanine peptides (2Nal^2^, 2Nal^3^, 2Nal^6^, 2Nal^9^, and 2Nal^10^), and β-cyclohexylalanine peptides (Cha^2^, Cha^3^, Cha^6^, Cha^9^, Cha^10^, and Cha^2^Cha^10^). The microdilution broth method was the technique used to determine antimicrobial assay against VRE (700 221), and the MICs values obtained for Anoplin and its 19 analogs varied from 0.8 to 170.6 µM. The positions 5- and 8-modified peptide, Phe^5^Trp^8^, showed the best MIC, 0.8 µM. This excellent activity is related to the increased hydrophobicity of the AMP and, consequently, the better interaction with the lipid membrane of bacteria. This study shows that small changes in peptide structure can alter the antimicrobial activity against certain bacteria.

The novel peptide BF2 was synthesized, purified, characterized, and biologically evaluated against clinical isolates of *VRE* by Singh et al. [102]. These authors analyzed the BF2 (RWRLLLLKKH) peptide against 2 ATCC strains (ATCC29212 and ATCC51299) and 26 clinical VRE isolates, clinical isolates from blood culture, wound swabs, the right abdomen, a hydrocathered tip, bile, and pus. The peptide showed antimicrobial activity against the standard strains with an MIC of 25 µg/mL and an MIC in the range of 6.25–12.5 µg/mL against the clinical isolates, which shows better activity against these strains. The hemolysis determined the cytotoxicity of BF2 peptide and showed that, up to 5MIC, it was not hemolytic. They also performed a synergistic study between BF2 and the antibiotics (vancomycin, teicoplanin, and linezolid) against ATCC51299 and clinical isolates; and the results showed that vancomycin with BF2 has a synergistic effect with a fractional inhibitory concentration (FIC) index of 0.06–0.25, against ATCC and resistant strains, respectively. Simultaneously, BF2 with teicoplanin showed an FIC of 0.02–0.27 for ATCC and clinical isolates. They observed no effect for the combination BF2/linezolid. An in vivo study was performed on Wistar albino female rats to verify the antimicrobial activity of the BF2 peptide studied by reducing the cell count in the blood of the animals. The inoculation of bacterial cells (VRE) was intravenous. One hour after administration, the animal groups were treated with 12.5 (MIC), 25 (2MIC), and 37.5 mg/kg (3MIC) of the peptide intravenously. After 1 h of treatment, the animals’ blood was collected and plated for quantifying the number of bacterial cells. After a two-day therapy, reductions of 73%, 76%, and 82% were observed for the antimicrobial peptide treatments at MIC, 2MIC, and 3MIC concentrations, respectively, in comparison with the negative control. In contrast, the positive control group, which received linezolid, had the number of colonies reduced by 93%. BF2 is a peptide with good antimicrobial activity against vancomycin-resistant clinical isolates. It is nontoxic and effective in vivo, making it a promising antimicrobial to be studied both alone and in combination with antibiotics.

A cationic peptide composed of 22 amino acids, named Pexiganan or MSI 78, was evaluated in vitro against VRE strains by Flamm et al. [103]. The strains were isolated from diabetic foot infections, and the antimicrobial activity of pexiganan was evaluated by the broth microdilution testing method. The peptide showed good activity against VRE strains, with MIC values of 4 or 8 µg/mL. The results showed that MSI 78 can be used as an antibiotic for VRE strains in diabetic patients.

Delpech et al. [104] evaluated the peptide AP-CECT7121 alone and combined with gentamicin against five clinical isolates of VRE, ampicillin, and tetracycline. The five (DF02–043, DF04–056, DF02–065, DF03–072, and DF03–078)-resistant strains were isolated from mastitic cows. The bactericidal activity was determined by the agar dilution method, and it was observed that the peptide AP-CECT121 was able to kill the isolates, reducing viable cells by more than 4.0log_10_ CFU/mL. However, a synergistic effect could not be observed when combined with gentamicin, as there was only a 0.2 to 0.9 log_10_ CFU/mL reduction in viable cells. Therefore, AP-CEC7121 showed good activity alone and can be considered a candidate to treat bovine mastitis caused by VRE.

To evaluate synergistic effects, Wu et al. [105] performed a check-board study with the antimicrobial peptide P-113 and its derivatives against VRE. The following derivatives were developed with phenylalanine-(Phe-P-113), β-naphthylalanine-(Nal-P-113), β-diphenylalanine-(Dip-P-113), and β-(4,4′-biphenyl)alanine-(Bip-P-113)-substituted histidine-rich antimicrobial peptide P-113 (Ac-AKRHHGYKRKFH-NH2). A clinically isolated strain of VRE (BCRC 15B0132) was purchased from Bioresources Collection & Research Center (BCRC, FIRDI, Hsinchu, Taiwan). The antimicrobial activity assay was determined and vancomycin, P-113, and Phe-P-113 exhibited MIC > 64 µg/mL against *E. faecium* BCRC 15B0132, while the derivatives Bip-P-113, Dip-P-113, and Nal-P-113 showed an MIC equal to 4 µg/mL. The synergic effect study was performed by combining vancomycin and peptides at the sub-inhibitory concentration (1/4 x MIC). The authors observed that Bip-P-113, Dip-P-113, and Nal-P-113 had a synergistic effect, and the other peptides showed no interaction. To evaluate the mechanism of resensitization of *Enterococcus faecium* to vancomycin by P-113 and its derivatives, the scientists experimented with BODIPY-labeled vancomycin. They noted that peptides with bulky unnatural amino acids increased vancomycin’s entry into the VRE with the order Bip-P-113 = Nal-P-113 > Dip-P-113 > Phe-P-113 = P-113. This study shows that P-113 derivatives can be used alone or in combination with vancomycin against VRE.

Wang et al. [106] discovered a new peptide, albopeptide, from the soil bacterium *Streptomyces albofaciens* (NCIMB 10975). After identifying this compound, the researchers synthesized its isomer and confirmed the peptide structure as l-Val-Dha-(E)-Dhb. With the low yield of the natural albopeptide for antibacterial activity against VRE, they used a 50:50 mixture of synthetic and natural peptide material for the experiment. The combination of peptides resulted in a potent activity against VRE K60-39, with a minimum inhibitory concentration (MIC) value of 2.98 ± 0.07 µM.

### 5.2. Clarithromycin-Resistant Helicobacter pylori

Jiang et al. [107] studied the efficacy of the peptide Cbf-K16 (with 30 amino acids), in vitro and in vivo, against clarithromycin- and amoxicillin-resistant *H. pylori* SS1. An antimicrobial activity assay was carried out in vitro according to the microdilution technique, and the minimum inhibitory concentration (MIC) and minimum bactericidal concentration (*MBC*) obtained were 16 and 32 µg/mL, respectively. Cbf-K16 was also evaluated against *H. pylori* SS1-infected cells, GES-1 (gastric epithelial cells), and the compound reduced the bacterial count in the supernatant and intracellular samples to 1.9 and 2.9-log_10_ units, respectively, in addition to protecting the epithelial cells. An antimicrobial activity assay in mice infected with drug-resistant *H. pylori* SS1 was performed with C57BL/6 male mice (4–6 weeks old). The animals received sodium bicarbonate by the intragastric route, and 0.3 mL (1 × 10^9^ CFU/mL) of clarithromycin-resistant *H. pylori* SS1 on three alternate days for infection development. After infection, the mice were divided into different groups: the control group, triple therapy (omeprazole, clarithromycin, and amoxicillin), and peptide Cbf-K16 treatments at different doses (5, 10, and 20 mg/kg). The rats were treated daily for two weeks orally by the intragastric route, and 24 h after the last administration, they were euthanized, their blood was collected, and their stomachs were removed for analysis. Cbf-K16 at a dose of 20 mg/kg reduced the bacterial count in the stomach tissues of the animals to 3.9-log_10_ units, in comparison with the control group. It also reduced the inflammation of the ulcer and gastric mucosa, as shown in the histological examination. By contrast, the triple therapy could not reduce the bacterial load of the resistant strain and had no action in gastritis. This study showed that Cbf-K16 peptide has in vitro and in vivo activity against a strain of a clinical isolate of Clarithromycin-resistant *H. pylori*.

### 5.3. Fluoroquinolone-Resistant Salmonella

Szabo et al. [108] studied antimicrobial peptides (AMPs) that carry interaction domains with both the bacterial membrane and intracellular target and exhibit stability in the face of proteolytic degradation. The peptide AE-APO with the sequence (H-Chex-Arg-Pro-AspLys-Pro-Arg-Pro-Tyr-Leu-Pro-Arg-Pro-Arg-Pro-Pro-Arg-Pro-ValArg)_2_-Dab-NH_2_ (where Chex is a 1-amino-cyclohexane carboxylic acid and Dab is 2,4-diamino-butyric acid) was analyzed in vitro against a clinical isolate of fluoroquinolone-resistant *Salmonella enterica* serovar *Typhimurium*. The in vitro efficacy was determined by conventional antibacterial growth inhibition assays [109], and the minimum inhibitory concentration (MIC) was 8-32 mg/L, that is, the peptide showed activity against the fluoroquinolone-resistant strain.

Cationic peptides LS-sarcotoxin and LS-stomoxyn were tested against a panel of 114 clinical MDR Gram-negative bacterial isolates, including fluoroquinolone-resistant *Salmonella enterica*. LS-sarcotoxin and LS-stomoxyn showed MIC_50_ and MIC_90_ values of 4 and 8 mg/L, respectively, against clinical isolate *S. enterica*. The cytotoxicity activity of the peptides was determined by measuring intracellular ATP levels of HepG2 cells, and the authors observed no cytotoxic effect, only an 80% viability at concentrations of 420 and 433 mg/L for LS-sarcotoxin and LS-stomoxyn, respectively. In the hemolytic assay, for both peptides, the minimal hemolytic concentration determined for human erythrocytes was 1024 mg/L, i.e., both showed no evidence of hemolytic activity. The in vivo tolerability assay was performed on healthy male Swiss mice. The animals received a single intravenous dose of 10 mg/kg of the respective peptides (LS-sarcotoxin and LS-stomoxyn). They were evaluated daily for one month and no signs of toxicity were observed for either of the administered compounds. The pharmacokinetic profile of the peptides was determined in this study at 0.08, 0.25, 0.5, 1, 2, 4, 8, and 24 h after administration of the 10 mg/kg dose of the respective compounds. Then, the 10 µL blood sample was collected to quantify the peptide in plasma by tandem mass spectrometry (LC-MS), and the plasma concentrations above the lower limit of quantification (LLOQ) for LS-sarcotoxin and LS-stomoxyn were 1000 and 250 ng/mL, at 5 and 15 min after administration, respectively. The peptides showed good antimicrobial activity against the resistant strain of *S. enterica* and no toxicity, and thus, they can be considered promising new antibiotics. However, they did not exhibit a good pharmacokinetic profile, which needs to be improved before in vivo efficacy assays can be performed [110].

### 5.4. Staphylococcus aureus

Various AMPs have shown potential against methicillin-resistant, vancomycin-resistant, and intermediate *S. aureus* (MRSA/VRSA/VISA), and most of the peptides against *S. aureus* were isolated from snake venom [111], bee venom [112], the skin of reptiles [113], and even from birds [114]. This is probably one of the most widely studied bacteria in recent decades of AMP applications.

A study that applied the combined action of melittin/mupirocin (4.4 and 14.22 μg/mL) demonstrated the synergistic effect between the AMP-drug during treatment and highlighted its application as anti-biofilm formers against eight strains of MRSA [115]. This AMP is capable of inserting itself into the lipid membrane, forming a toroidal pore, and it also has an anti-VRE potential effect [116]. Likewise, WLBU2 is a promising peptide evaluated with CPB and HPB, which showed activity even in the middle of the biofilm formed. These results allow us to have a base molecule, from which to generate new homologs that reduce its cytotoxicity to take it to the next experimental stage [117]. The previously described Bip-P-113 also showed promising activity against VISA, and thus, it created expectations about the synergy of peptidomimetics and AMPs [105]. For other high-priority bacterial strains, no AMP studies have been reported yet.

## 6. Challenges and Perspectives

Many AMPs, as demonstrated, have excellent properties against resistant bacteria. The difficulty in being applied to biological systems lies in the degradation and denaturation caused by biomolecules, such as proteolytic enzymes, and the gastric environment, as peptides and proteins can be hydrolyzed during their transport through the gastrointestinal tract when they are administered mainly orally [118]. This challenge has been deeply investigated through the use of protection and drug delivery technologies, such as micro and nanotechnology, with promising results for the development of new drugs [119,120].

Some AMPs present hemolytic activity, which is considered an important factor to be controlled and evaluated, as human cells could be lysed before the AMPs fulfill their objective [121]. However, some advances indicate that the manipulation of hydrophobicity could be crucial for this type of molecule, and for this reason, more studies of amino acid substitution within the peptide sequence must be carried out [56]. Furthermore, a previous study by our research group indicated that the use of microencapsulation with alginate by ionic gelation could decrease the hemolytic activity without affecting the antimicrobial activity of the Ctx(Ile^21^) peptide, making it useful as an oral food additive in formulations against MDR bacteria [5]. To the best of our knowledge, this is the first review article including AMPs against bacteria from the WHO priority list.

## 7. Conclusions

The progress of studies on AMPs against bacteria of critical and high priority according to the WHO has shown that the effectiveness of these peptides is, in certain cases, even better than that of conventional drugs. In addition, the combined use of AMP and drugs has a synergistic effect capable of reactivating the antimicrobial action of many obsolete drugs. The continuous study of AMPs in the pharmaceutical industry has produced promising results to solve the great problem of current bacterial resistance.

## Figures and Tables

**Figure 1 pharmaceutics-13-00773-f001:**
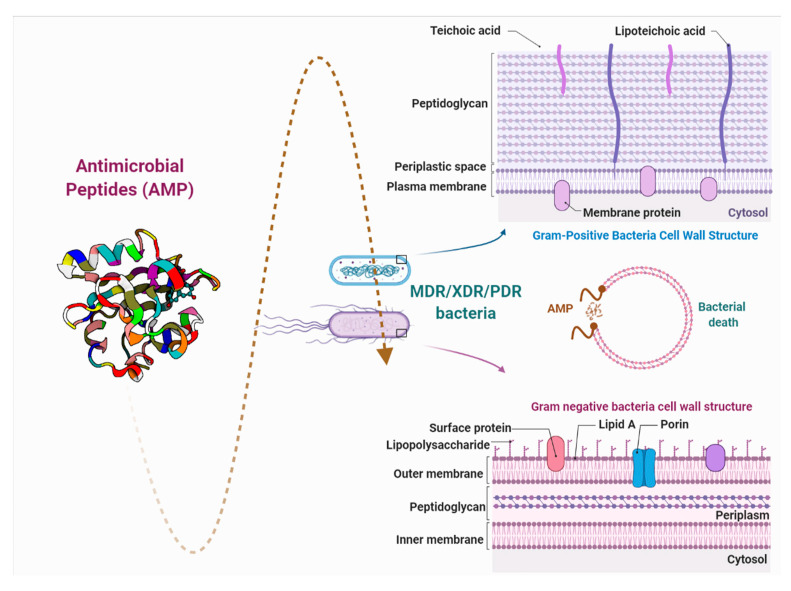
Brief description of the interaction between antimicrobial peptides and the bacterial cell wall. PDR: pandrug resistance, XDR: extensive drug resistance, and MDR: multidrug resistance.

**Figure 2 pharmaceutics-13-00773-f002:**
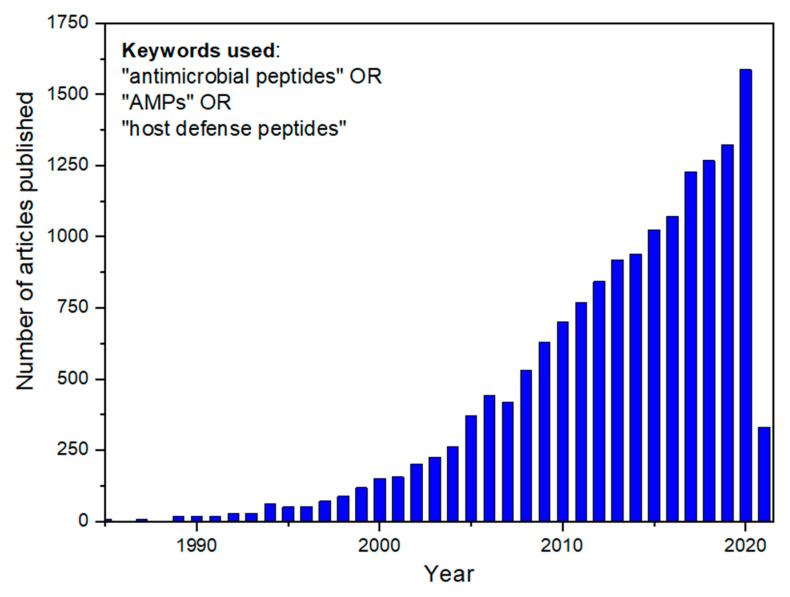
Number of scientific articles published in academic journals since 1985. The inset shows the keywords employed to perform the search. For the 2021 year, showing the number of articles published until the date of data search (3 March 2021), the number of papers (330) is higher than that corresponding to the whole of 2004.

**Table 1 pharmaceutics-13-00773-t001:** New FDA-approved drugs; detailed review 2017–2021 [24].

Drug	Approval Date	FDA-Approved Application	Mechanism of Action	Antibacterial Activity
cefiderocol	14 November 2019	To treat patients with complicated urinary tract infections who have limited or no alternative treatment options	- Cephalosporin with activity against Gram-negative aerobic bacteria. - Functions as a siderophore and binds to extracellular free ferric iron. - Passive diffusion via porin channels. - Actively transported across the outer cell membrane of bacteria into the periplasmic space using a siderophore iron uptake mechanism. Cefiderocol exerts bactericidal action by inhibiting cell wall biosynthesis through binding to penicillin-binding proteins (PBPs).	Gram-negative: *Escherichia coli*, *Enterobacter cloacae complex*, *Klebsiella pneumoniae*, *Proteus mirabilis*, *Pseudomonas aeruginosa*, *Acinetobacter baumannii*, *Citrobacter freundii complex*, *Citrobacter koseri*, *Klebsiella aerogenes*, *Klebsiella oxytoca*, *Morganella morganii*, *Proteus vulgaris*, *Providencia rettgeri*, *Serratia marcescens*, *Stenotrophomonas maltophilia*.
imipenem, cilastatin, and relebactam	16 July 2019	To treat complicated urinary tract and complicated intra-abdominal infections	- Imipenem is a penem antibacterial drug, cilastatin sodium is a renal dehydropeptidase inhibitor, and relebactam is a beta lactamase inhibitor. - Cilastatin limits the renal metabolism of imipenem and does not have antibacterial activity. The bactericidal activity of imipenem results from the binding to PBP 2 and PBP 1B in Enterobacteriaceae and *Pseudomonas aeruginosa* and the subsequent inhibition of PBPs. Inhibition of PBPs leads to the disruption of bacterial cell wall synthesis. Imipenem is stable in the presence of some beta lactamases. - Relebactam has no intrinsic antibacterial activity, and it protects imipenem from degradation by certain serine beta lactamases such as Sulhydryl Variable (SHV), Temoneira (TEM), and Cefotaximase-Munich.	Complicated Urinary Tract Infections and Complicated Intra-abdominal Infections. Some important bacters: *Citrobacter freundii*, *Klebsiella aerogenes*, *Enterobacter cloacae*, *Escherichia coli*, *Klebsiella oxytoca*, *Klebsiella pneumoniae*, *Pseudomonas aeruginosa*, *Bacteroides caccae*, *Bacteroides fragilis*, *Bacteroides ovatus*, *Bacteroides stercoris*, *Bacteroides thetaiotaomicron*, *Bacteroides uniformis*, *Bacteroides vulgatus*, *Fusobacterium nucleatum*, *Parabacteroides distasonis*. *Enterococcus faecalis*, Methicillin-susceptible *Staphylococcus aureus*, *Streptococcus anginosus*, *Streptococcus constellatus*. *Citrobacter koseri*, *Enterobacter asburiae*, etc.
lefamulin	19 August 2019	To treat adults with community-acquired bacterial pneumonia	- Systemic pleuromutilin antibacterial. - Inhibits bacterial protein synthesis through interactions (hydrogen bonds, hydrophobic interactions, and van der Waals forces) with the A- and P-sites of the peptidyl transferase center (PTC) in domain V of the 23s rRNA of the 50S subunit. The binding pocket of the bacterial ribosome closes around the mutilin core for an induced fit that prevents the correct positioning of tRNA.	*S. pneumoniae*, *H. Influenzae*, and *M. pneumoniae* (including macrolide-resistant strains), and bacteriostatic against *S. aureus*, and *S. pyogenes* at clinically relevant concentrations
pretomanid	14 August 2019	For treatment-resistant forms of tuberculosis that affect the lungs	- Nitroimidazooxazine antimycobacterial drug. - inhibiting mycolic acid biosynthesis, thereby blocking cell wall production.	Mutations in five *M. tuberculosis* genes (ddn, fgd1, fbiA, fbiB, and fbiC) have been associated with pretomanid resistance.
omadacycline	2 October 2018	To treat community-acquired bacterial pneumonia and acute bacterial skin and skin structure infections	- Aminomethylcycline antibacterial (tetracycline class of antibacterial drugs). - The drug binds to the 30S ribosomal subunit and blocks protein synthesis. - Active in vitro against Gram-positive bacteria expressing tetracycline resistance active efflux pumps (tetK and tet L) and ribosomal protection proteins (tet M). In general, omadacycline is considered bacteriostatic; however, omadacycline has demonstrated bactericidal activity against some isolates of *S. pneumoniae* and *H. influenzae*.	Gram-positive bacteria that carried ribosomal protection genes (tet M) and efflux genes (tet K and tet L), and in *Enterobactericeae* that carried the tetB efflux gene. Some *S. aureus*, *S. pneumoniae*, and *H. influenzae* strains carrying macrolide resistance genes (erm A, B, and/or C), or ciprofloxacin resistance genes (gyrA and parC) and beta-lactamase-positive *H. influenzae*.
eravacycline	27 August 2018	To treat complicated intra-abdominal infections in patients 18 years of age and older	- Fluorocycline antibacterial (tetracycline class of antibacterial drugs). - This drug disrupts bacterial protein synthesis by binding to the 30S ribosomal subunit, thus preventing the incorporation of amino acid residues into elongating peptide chains.	In general, is bacteriostatic against Gram-positive bacteria (e.g., *Staphylococcus aureus* and *Enterococcus faecalis*); however, in vitro bactericidal activity has been demonstrated against certain strains of *Escherichia coli* and *K. pneumoniae*.
plazomicin	25 June 2018	To treat adults with complicated urinary tract infections	- Aminoglycoside that acts by binding to the bacterial 30S ribosomal subunit, thereby inhibiting protein synthesis.	*Enterobacteriaceae* in the presence of certain beta-lactamases, including extended-spectrum beta-lactamases (TEM, SHV, CTX-M, AmpC), serine carbapenemases (KPC-2, KPC-3), and oxacillinase (OXA-48). Bacteria producing metallo-beta-lactamases often co-express 16S rRNA methyltransferase, conferring resistance to plazomicin.
secnidazole	15 September 2017	To treat bacterial vaginosis	- 5-nitroimidazole antimicrobial. - 5-nitroimidazoles enter the bacterial cell as an inactive prodrug where the nitro group is reduced by bacterial enzymes to radical anions. It is believed that these radical anions interfere with the bacterial DNA synthesis of susceptible isolates.	*Bacteroides* spp., *Gardnerella vaginalis*, *Prevotella* spp., *Mobiluncus* spp., *Megasphaera-like* type I/II
meropenem and vaborbactam	29 August 2017	To treat adults with complicated urinary tract infections	- The meropenem is a penem antibacterial drug. - The bactericidal action of meropenem results from the inhibition of cell wall synthesis. Meropenem penetrates the cell wall of most Gram-positive and Gram-negative bacteria to bind PBP targets. Meropenem is stable to hydrolysis by most beta-lactamases, including penicillinases and cephalosporinases produced by Gram-negative and Gram-positive bacteria, with the exception of carbapenem hydrolyzing beta-lactamases. The vaborbactam is a nonsuicidal beta-lactamase inhibitor that protects meropenem from degradation by certain serine beta-lactamases such as *Klebsiella pneumoniae* carbapenemase (KPC). Vaborbactam does not have any antibacterial activity. Vaborbactam does not decrease the activity of meropenem against meropenem-susceptible organisms.	Gram-negative bacteria: *Enterobacter cloacae* species complex, *Escherichia coli*, *Klebsiella pneumoniae*, *Citrobacter freundii*, *Citrobacter koseri*, *Enterobacter aerogenes*, *Klebsiella oxytoca*, *Morganella morganii*, *Proteus mirabilis*, *Providencia* spp., *Pseudomonas aeruginosa*, *Serratia marcescens*.
delafloxacin	19 June 2017	To treat patients with acute bacterial skin infections	- Fluoroquinolone class of antibacterial drugs and is anionic in nature. - Inhibition of both bacterial topoisomerase IV and DNA gyrase (topoisomerase II) enzymes, which are required for bacterial DNA replication, transcription, repair, and recombination.	Gram-positive bacteria *Staphylococcus aureus* (including methicillin-resistant and methicillin-sensitive strains), *Staphylococcus haemolyticus*, *Staphylococcus lugdunensis*, *Streptococcus pyogenes*, *Streptococcus agalactiae*, *Streptococcus anginosus* Group (including *S. anginosus*, *S. intermedius*, and *S. constellatus*), *Enterococcus faecalis*, *Streptococcus dysgalactiae*. Gram-negative bacteria *E. coli*, *K. pneumoniae*, *Enterobacter cloacae*, *P. aeruginosa*, *Enterobacter aerogenes*, *Haemophilus parainfluenzae*, *Klebsiella oxytoca*, *Proteus mirabilis*.

**Table 2 pharmaceutics-13-00773-t002:** Classification of antimicrobial agents according to the Food and Drug Administration (FDA) [25].

	Group	Sub-Group	Classification	Main Drugs
**Antimicrobial agents**	ß-Lactamic	Penicillins	natural penicillins or benzilpenicillins	crystalline penicillin
				penicillin G procaine
				benzathine penicillin G
				penicillin V
			aminopenicillins	ampicillin
				amoxicillin
			penicillins resistant to penicillinases	oxacillin
				methicillin
				carbenicillin
				ticarcillin
				piperacillin
			Broad-spectrum penicillins	carboxypenicillins (carbenicillin and ticarcillin)
				ureido-penicillins (mezlocillin, piperacillin, and azlocillin)
		Cephalosporins	1st generation	cephalothin
				cefazolin
				cephalexin
				cefadroxil
			2nd generation	cefoxitin
				cefuroxime
				cefaclor
			3rd generation	cefotaxime
				ceftriaxone
				ceftazidime
			4th generation	cefepime
		Carbapenems	-	Imipenem
				meropenem
				ertapenem
				doripenem
				biapenem
				tebipenem
		Monobactams	-	aztreonem
	Quinolones	-	-	levofloxacin
				gatifloxacin
				moxifloxacin
				gemifloxacin
	Glycopeptides	-	-	vancomycin
				teicoplanin
				branchplanin
	Oxazolidinones	-	-	linezolid
	Aminoglycosides	-	-	streptomycin
				gentamicin
				tobramycin
				amikacin
				netilmicin
				paromomycin
				spectinomycin
	Macrolides	-	-	azithromycin
				clarithromycin
				erythromycin
				spiramycin
				myocamycin
				roxithromycin
	Lincosamines	-	-	lincomycin
	Nitroimidazole	-	-	metronidazole
	Chloramphenicol	-	-	chloramphenicol
	Streptogramins	-	-	quinupristin
				dalfopristin
	Sulfonamides	-	-	sulfanilamide
				sulfisoxazole
				sulfacetamide
				para-aminobenzoic acid
				sulfadiazine and sulfamethoxazole
	Tetracyclines	-	-	tetracyclines
	New Antimicrobials	Glycylcyclines	-	tigecycline
		Polymyxins	-	colistin (polymyxin E)
				polymyxin B
		Daptomycin	-	daptomycin

**Table 3 pharmaceutics-13-00773-t003:** Several reviews (2019 to 2021) of antimicrobial peptides applied against WHO-list (CPB).

AMPs	Peptide Sequence	MIC (μg/mL)	System	Natural Source	PDR/MDR/XDR Bacteria	Ref.
ZY4 peptide	VCKRWKKWKRKWKKWCV In the sequence, disulfide bond (C-C) is formed by the Cystein.	2.0–4.5	In vitro/In vivo	Snake venom of *Bungarus fasciatus*	*P. aeruginosa* (CICC21625, CMCC10104, C1, C2, C3, and C5)	[49]
		4.6–9.4			*A. baumannii* (22933, CN40, 18C116, 18C132, 18C135, and 18C136)	
Zp3	GIIAGIIIKIKK	4 µM	In vitro/In vivo	NR	*A. baumannii* ATCC 19606	[60]
WWW Symmetric peptides	XRWWWRX (XRW), XWRWWWRWX (XWRW), XRRWWWRRX (XRRW) X = G, I, L, W, F, V, A	2–>128 μM	In vitro/In vivo	NR	*E. coli* ATCC 25922, *P. aeruginosa* ATCC 27853/9027, *Acinetobacter baumannii* ATCC 19606, *K. pneumoniae* ATCC 700603, *E. coli* ML-35 ATCC 43837, *E. coli* 780/804, *P. aeruginosa* 50/73, *A. baumannii* 9931/97830	[61]
NR	HKEMPFPK, TTMPLW, YYQQKPVA, and AVPYPQR	1.5–5 mg/mL	In vitro	Casein prediction	*A. baumannii* ATCC 19606, *E. coli* O157 NCTC 12900, *P. aeruginosa* PAO1	[62]
B1CTcu5	LIAGLAANFLPQILCKIARKC	12.5	In vitro	*Clinotarsus curtipes*	*M. tuberculosis*	[56]
Ctx(Ile^21^)-Ha	GWLDVAKKIGKAAFSVAKSFI	64 µM	In vitro	*Hypsiboas albopunctatus*	*A. baumannii* MDR	[5]
					*P. aeruginosa* MDR	
CDP-B11 peptide	VRNSQSCRRNKGICVPIRCPGSMRQIGTCLGAQVKCCRRK	25	In vitro	Cow	*A. baumannii* # 0035	[63]
		100			*E. coli* # 0061	
		200			*E. coli* # 0346, *P. aeruginosa* # 0054 y *K. pneumoniae* # 0347	
P10 + conventional antibiotics	LAREYKKIVEKLKRWLRQVLRTLR	4–32		LL-37 derivative	*A. baumannii* XDR 1, XDR 2, XDR 3, XDR 4, XDR 5, ATCC 19606.	[64]
		8–16			*P. aeruginosa* colistin resistant 1,2,3,4,5, and ATCC 27853.	
K11	KWKSFIKKLTKKFLHSAKKF	NR	In silico/In vitro	Cecropin A1, melittin, and magainin	*A. baumannii*, *P. aeruginosa*, and *K. pneumoniae*	[65]
Magainin 2 (Mag2) (molecule and homologues)	GIGKFLHSAKKFGKAFVGEIMNS	12.5	In vitro	African clawed frog	*P. aeruginosa*	[66]
	GIKKFLKSXKKFVKXFK	3125				
	**S5**IKK**S5**LKSAKKFVKAFK	1.56				
	GIKK**S5**LKS**S5**KKFVKAFK	0.78				
	GIKKFLK**S5**AKK**S5**VKAFK	0.78				
	GIKKFLKS**S5**KKF**S5**KAFK	3125				
	GIKKFLKSAKK**S5**VKA**S5**K	0.78				
	GIKK**R8**LKSAKK**S5**VKAFK	1.56				
	GIKKFLK**R8**AKKFVK**S5**FK	3125				
	GIKKFLKS**R8**KKFVKA**S5**K	3125				
Phe-Lys-Phe tripeptide	FKF	>45.4 µM	In vitro	NR	*K. pneumoniae* ATCC 13883	[67]
					*K. pneumoniae* ATCC 700603	
		45.4 µM			*E. coli* ESBL 76	
					*E. coli* ESBL 63	
		34.0 µM			*A. baumannii* 5025055	
					*P. aeruginosa* 760111736	
					*E. coli* ATCC 25922	
		NR	In vivo		*P. aeruginosa* ATCC 27853	
Cec4	GWLKKIGKKIERVGQNTRD ATIQAIGVAQQAANVAATLKGK	4	In vitro	NR	200 biofilm-forming strains MDR and XDR of *P. aeruginosa* (clinic isolates)	[68,69]
WLBU2	RRWVRRVRRWVRRVVRVVRRWVRR	1.5–3 µM	In vivo/In vitro	eCAP modified	24 biofilm-forming strains MDR and XDR of *A. baumannii*	[70]
		42,500			*K. pneumoniae* BAA-2146 and 700603	[71]
		10,625	In vitro		*A. baumannii* BAA-1605	
BSI-9 analogs	K-Nal-KK-Bip-O2Oc-Nal-KS	1–32	In vitro	Cycllic peptide	*S. aureus* ATCC 29213 (SA)	[72]
		2–32			*P. aeruginosa* ATCC 27853	
		32–>64			*E. coli* ATCC 29522	
					*A. baumannii* ATCC 19606	
TC19	LRCMCIKWWSGKHPK	3.75 µM	In vitro	Thrombocidin-1- human peptide-derived	*E. coli* ESBL	[73]
			In vivo		*A. baumannii* LUH15100 and *S. aureus* JAR060131	
PepC/PepW	ILIACLGLKLLRYRRIY/WWWA{Dab}YGL{Dab}LL{Dab}Y{Dab}{Dab}WY	8 and 1 µM	In vivo/In vitro	NR	*E. coli* W3110	[55]
		2 µM			*A. baumannii* 5075	
		2 and 1 µM			*A. baumannii* 17978	
		>128 and 2 µM			*K. pneumoniae* MKP103	
					*K. pneumoniae* ATCC 1705	
					*K. pneumoniae* clinical isolate 1433	
					*K. pneumoniae* clinical isolate 1434	
					*K. pneumoniae* ATCC 43816 (K2 serotype/hypermucoviscous)	
		64 and 2 µM			*K. pneumoniae* ATCC 13883 (K3 serotype)	
Human β-defensin 2	HBD2/L-HBD2	0.25–0.5 μM,	In vitro	Human	Pyorubin-producing *P. aeruginosa* strain	[74]
Human β-defensin 3	HBD3	1 μM			*A. baumannii* ATCC 19606	
Tridecaptin M + rifampicin	Gdab*GS*W*SDabDab*IQIαI*S–D-aminoacids*	16	In vitro	*Paenibacillus* sp. M152		[75]
Tilapia Piscidin 4 (TP4)	FIHHIIGGLFSAGKAIHRLIRRRRR	0.52	In vivo/In vitro	Nile Tilapia (*Oreochromis niloticus*)	*P. aeruginosa* (ATCC 19660)	[76]
		3125			*K. pneumoniae* (YT32)	
		<1.56			*E. coli* (YT39)	
					*A. baumannii* (Icu53)	
					*A. baumannii* (Sk44)	
		3127			*K. pneumoniae* (NDM-1)	
Oncocin	VDKPPYLPRPRPPRRIYNR	8 μM	In vitro	*Oncopeltus fasciatus* (milkweed bug)	*E. coli* ATCC 25922	[77]
		4 μM			*K. pneumoniae* ATCC 10031	
		2 μM			*K. pneumoniae* CMCC 46117	
		16 μM			*E. coli* ATCC 25922 (ΔSbmA)	
MDAP-2	SRDSRPVQPRVQPPPPPKQKPSIYDTPIRRPGGRKTMYA	128 μM	In vitro	*Musca domestica larvae*	*E. coli* ATCC 25922	
		512 μM			*K. pneumoniae* ATCC 10031	
					*K. pneumoniae* CMCC 46117	
		256 μM			*E. coli* ATCC 25922 (ΔSbmA)	
OM19R (MDAP-2 + Oncocin)	VDKPPYLPRPR PIRRPGGR	1 μM	In vitro	hybrid	*E. coli* ATCC 25922	
		256 μM			*K. pneumoniae* ATCC 10031	
					*K. pneumoniae* CMCC 46117	
					*E. coli* ATCC 25922 (ΔSbmA)	
Piscidin 3 (g6498.t1)	CIMKHLRNLWNGAKAIYNGAKAGWTEFK	45	In vivo/In vitro	*Oreochromis niloticus*	*P. aeruginosa*	[78]
Piscidin 1 homologues	I9A-piscidin-1	3.1	In vitro		Clinical strain of colistin-resistant *A. baumannii*	[79]
	I16A-piscidin-1					
	I9K-piscidin-1					
	I16K-piscidin-1					
EcDBS1R6	PMKKLFKLLARIAVKIPVW	3 μM	In vitro/In silico	NR	*E. coli* ATCC 25922	[80]
		6.25 μM			*P. aeruginosa* ATCC 27853	
		3 μM			*K. pneumoniae* ATCC 13883	
		3 μM			*A. baumannii* ATCC 19606	
Iztli peptide 1 (IP-1)	KFLNRFWHWLQLKPGQPMY	8 μg/100 μL	In vitro	NR	*M. tuberculosis* H37Rv (ATCC 27294)	[81]
Buforin II + monocarboxylic acid derivative	TRSSRAGLQFPVGRVHRLLRK	1 μM	In vitro	NR	*E. coli* (MG 1655)	[82]
		2 μM			*P. aeruginosa* (AR 0229)	
		0.5 μM			*E. coli* (AR 0114)	
					*A. baumannii* (Naval-17)	
		4 μM			*K. pneumoniae* (AR 0113)	
CAMP-CecD	RNFFKRIRRAGKRIRKAI	32	In vitro/In silico	Cecropin D-Derived from insects	*K. pneumoniae* (wild-type KP)	[83]
		256			*K. pneumoniae* (MDR-KP)	
		128			*E. coli* ATCC 25922	
		256			*K. pneumoniae* (2146)	
		64			*P. aeruginosa* (wild-type PA)	
		32			*P. aeruginosa* (MDR-PA)	
		32			*P. aeruginosa* (27853)	
SET-M33 protease resistant	(KKIRVRLSA)4K2KβA-OH	0.7–3 μM	In vivo/In vitro	NR	*S. aureus* (6 strains MDR/XDR)	[84]
					*P. aeruginosa* (7 strains MDR/XDR)	
		3–6 μM			*A. baumannii* (3 strains MDR/XDR)	
					*E. coli* (8 strains MDR/XDR)	
					*K. pneumoniae* (5 strains MDR/XDR)	
DP7	VQWRIRVAVIRK	16–32	In vivo/In vitro/In silico	NR	*P. aeruginosa* (MDR various strains)	[85]
DP7 + vancomycin or azithromycin		NR			*S. aureus*, *P. aeruginosa*, *A. baumannii*, and *E. coli*	
DP-C and DP-C Dimer	[(KLAKLAK)*2-L-C] *D-amino acids (right-handed).	16–32	In vitro/In silico	NR	*P. aeruginosa*	[86]
		4–8			*A. baumannii*	
		4–16			*E. coli*	
		16–>128			*K. pneumoniae*	
Ω76 peptide	FLKAIKKFGKEFKKIGAKLK	16	In vivo/In vitro	Mag derivative	carbapenem and tigecycline-resistant *A. baumannii*	[87]
		128	In vitro		*K. pneumoniae*	
		4			*E. coli*	
NuriPep 1653	VRGLAPKKSLWPFGGPFKSPFN	12	In vitro	P54 nutrient reservoir protein of *Pisum sativum*	*A. baumannii*-resistant (ColRAB) PDR	[88]
		12			*A. baumannii* susceptible (colSAB)	
		400			*K. pneumoniae* 50183	
		100			*E. coli* ATCC 25922	
		8			*P. aeruginosa* ATCC 27853	
Tachyplesin III	KWCFRVCYRGICYRKCR	10 mg/kg	In vivo	β-sheet AMP	*P. aeruginosa* 1409	[89]
					*A. baumannii* 1409	
Melittin	NR	16	In vivo/In vitro	NR	*A. baumannii* (XDR and MDR strains, clinic isolates)	[90]
Protegrin-1	RGGRLCYCRRRFCVCVGR	4–8	In vitro	β-sheet peptides (cathelicidin family)	*A. baumannii* (XDR and MDR strains from surgical wounds)	[91]
Aurein 1.2	GLFDIIKKIAESF-NH2	16	In vitro	NR	*A. baumannii* ATCC 19606	[92]
CAMEL	KWKLFKKIGAVLKVL-NH2	2				
Citropin 1.1	GLFDVIKKVASVIGGL-NH2	16				
LL-37	LLGDFFRKSKEKIGKEFKRIVQRIKDFLRNLVPRTES	4				
Omiganan	ILRWPWWPWRRK-NH2	32				
r-Omiganan	KRRWPWWPWRLI-NH2	16				
Pexiganan	GIGKFLKKAKKFGKAFVKILKK-NH2	2				
Temporin A	FLPLIGRVLSGIL-NH2	128				
Bac5(1–17) native/Bac5-derivatives	wt, 258, 272, 278, 281, and 291 peptides	2–16	In vitro	NR	*E. coli* BW25113	[93]
					*E. coli* C679-12 EAEC:0104	
		4–>64			*A. baumannii* ATCC 19606	
		1–16			*E. coli* EURL-VTEC A07 EPEC:O111	
					*E. coli* SSI-OO15 EIEC	
		1–2			*E. coli* SSI-NN14 ETEC	
					*E. coli* EA22 ETEC	
		16–>64			*K. pneumoniae* ATCC 700603	
					*P. aeruginosa* ATCC 27853	
		8–>64			*E. coli* EURL-VTEC C07 STEC:O157	
					*E. coli* BW25113ΔsbmA	
SA4 peptide	IOWAGOLFOLFO-NH2	50	In vitro	NR	*A. baumannii* (MDR1, MDR2, MDR3, and MDR4)	[94]
SPO peptoid	nInOnWnAnGnOnLnFnOnLnFnO-NH2					
D87(Lys1–6 Arg-1)	Complex sequence (see reference)	0.8	In vitro	NR	7 Strains of *A. baumannii* resistant to Polymyxin B and Colistin and 20 Worldwide 2016 and 2017 Isolates resistant to 18 Classical Antibiotics	[95]
D84(Lys1–6 Lys-1)		0.5–0.7				
D85(Lys1–6 Orn-1)		0.5				
D86(Lys1–6 Dab-1)		1.0				
D105(Lys1–6 Dap-1)		1–1.2				
D101(Lys1Ser26-5 Lys-1)		0.8–1				
D102(Lys1Ser26-5 Dab-1)		0.6–0.7				
D85(K13A/K16A)-(Lys1–6 Orn-1)		1.9–2				
D86(K13A/K16A)-(Lys1–6 Dab-1)		0.9				
D105(K13A/K16A)-(Lys1–6 Dap-1)		1.3–2				
Agelaia-MPI	INWLKLGKAIIDAL	6.25–25 µM	In vitro	*Agelaia pallipes pallipes*	Clinical isolates of MDR *A. baumannii* identified as AB 02, AB 53, and AB 72	[96]
Polybia-MPII	INWLKLGKMVIDAL	12.5–25 µM		*Pseudopolybia vespiceps testacea*		
Con10	FWSFLVKAASKILPSLIGGGDDNKSSS			*Opisthacanthus cayaporum*		
NDBP-5.8	GILGKIWEGVKSLI	>25 µM		*Opisthacanthus cayaporum*		
Polydim-I	AVAGEKLWLLPHLLKMLLTPTP			*Polybia dimorpha*		
ARV-1502	Chex-RPDKPRPTLPRPRPPRPVR	50	In vivo	Insect	MDR *A. baumannii* Strain HUMC1	[97]
Bac7(1–35)	NR	4	In vitro	NR	*A. baumannii* XDR 420	[98]
					*A. baumannii* MDR 7B	
		2			*A. baumannii* MDR AB5075	
					*A. baumannii* MDR 215B	
P5 + meropenem	RIVQRIKKWLLKWKKLGY	16	In vitro	NR	*P. aeruginosa* M1351	[50]
PEP01 to PEP04	GKIMYILTKKS, FGIKLRSVWKK, FGIKLRSVWKR, and FGIKLRKVWKD	62.5–125	In silico	NR	*K. pneumoniae* MTCC619	[99]
NZX	GFGCNGPWSEDDIQCHNHCKSIKGYKGGYCARGGFVCKCY Disulfide bonds at position C4–C30, C15–C37, C19–C39	1.6–3.2 µM	In vitro	Plectasin derivate	*M. tuberculosis*	[100]

## Data Availability

Not applicable.

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
