# Peer review of "Challenge in the Discovery of New Drugs: Antimicrobial Peptides against WHO-List of Critical and High-Priority Bacteria"

_pharmaceutics, 2021, doi:10.3390/pharmaceutics13060773_

Round 1

Reviewer 1 Report

In this study, the author reviewed and analyzed the antimicrobial peptides against Critical Priority Bacteria (CPB) and High Priority Bacteria (HPB) that WHO listed. The conventional antimicrobial drugs were classified and listed, including β-Lactamic, Quinolones, and New Antimicrobials such as glycylcyclines, polymyxins, and so on. In this list, the concept of bacteria resistance (e.g., MDR, XDR, and PDR) was further discussion. Moreover, the designed AMP and its derivatives with strong antimicrobial activities against WHO-listed CPB were systematic analyzed in the manuscript. The peptide design and bioassays were also discussed in this manuscript. However, how to overcome the challenge of AMPs should be more elucidated in the manuscript, including poor stability by proteolytic degradation, high cytotoxicity/hemolytic activity.

Overall, the manuscript is suitable for publication in Pharmaceutics after a major revision.

Major comments

  1. Several reviews of antimicrobial peptides were shown in Table 2. Most of them exhibited strong antimicrobial activities, but ways to overcome the cytotoxicity or hemolytic activities were less discussed. How to design a kind of AMPs with lower toxicity? What is the relationship between the activity and toxicity of these AMPs?
  2. (page 13, line 330) The 3xMIC dosage of the BF2 peptide should be 37.5 mg/kg.
  3. (page 13, line 366) The sentence “…peptides s no interaction.” should be “…peptides showed no interaction.”
  4. The word sizes and fonts in the manuscript should be consistent.

Author Response

The authors are thankful for all comments, which will contribute to improve the quality of the attached revised version. All reviewer’s concerns will be discussed point-by-point, as described below:

Rebuttal for Editor and Reviewer

Reviewer 1:

In this study, the author reviewed and analyzed the antimicrobial peptides against Critical Priority Bacteria (CPB) and High Priority Bacteria (HPB) that WHO listed. The conventional antimicrobial drugs were classified and listed, including β-Lactamic, Quinolones, and New Antimicrobials such as glycylcyclines, polymyxins, and so on. In this list, the concept of bacteria resistance (e.g., MDR, XDR, and PDR) was further discussion. Moreover, the designed AMP and its derivatives with strong antimicrobial activities against WHO-listed CPB were systematic analyzed in the manuscript. The peptide design and bioassays were also discussed in this manuscript. However, how to overcome the challenge of AMPs should be more elucidated in the manuscript, including poor stability by proteolytic degradation, high cytotoxicity/hemolytic activity.

Overall, the manuscript is suitable for publication in Pharmaceutics after a major revision.

Major comments

  1. Several reviews of antimicrobial peptides were shown in Table 2. Most of them exhibited strong antimicrobial activities, but ways to overcome the cytotoxicity or hemolytic activities were less discussed. How to design a kind of AMPs with lower toxicity? What is the relationship between the activity and toxicity of these AMPs?

The "Challenges and perspectives" section was added.

  1. (page 13, line 330) The 3xMIC dosage of the BF2 peptide should be 37.5 mg/kg.

Corrected.

  1. (page 13, line 366) The sentence “…peptides s no interaction.” should be “…peptides showed no interaction.”

Corrected.

  1. The word sizes and fonts in the manuscript should be consistent.

Corrected and the English was corrected by a native speaker.

Reviewer 2 Report

I consider that in the Introduction section should be added a paragraph about the antibiotic resistance as a serious public health issue.

I consider that the data in Table I are very well-known and it would be more useful a table with the new FDA-approved antibiotics mentioned in the article and their spectrum of action.

The names of the microorganisms should be written using the Italic Style. Please check it in the entire manuscript. Please also check if the names are correctly written (e.g. Klebsiella pneumonia). Do check the font used, as from time to time looks different.

English language should be reviewed throughout the article, including the tables. Some paragraphs are difficult to follow due to grammar mistakes. There are many phrases that should be rephrased (e.g  “On the other hand, antimicrobial peptides (AMPs) are biomolecules present in all 38 living organisms and that are produced as defense mechanisms of the host” or “many of the diseases with drug broad resistance”). Do check all.

Author Response

The authors are thankful for all comments, which will contribute to improve the quality of the attached revised version. All reviewer’s concerns will be discussed point-by-point, as described below:

Rebuttal for Editor and Reviewer

Reviewer 2:

I consider that in the Introduction section should be added a paragraph about the antibiotic resistance as a serious public health issue.

A paragraph was added on antibiotic resistance as a serious public health problem.

I consider that the data in Table I are very well-known and it would be more useful a table with the new FDA-approved antibiotics mentioned in the article and their spectrum of action.

Being a review article, it is necessary to present Table 1 as it helps to better understand when resistance in CPB and HPB is explained. However, in-text information on new FDA-approved drugs has been expanded (new table was added).

The names of the microorganisms should be written using the Italic Style. Please check it in the entire manuscript. Please also check if the names are correctly written (e.g. Klebsiella pneumonia). Do check the font used, as from time to time looks different.

It has been corrected throughout the text.

English language should be reviewed throughout the article, including the tables. Some paragraphs are difficult to follow due to grammar mistakes. There are many phrases that should be rephrased (e.g  “On the other hand, antimicrobial peptides (AMPs) are biomolecules present in all 38 living organisms and that are produced as defense mechanisms of the host” or “many of the diseases with drug broad resistance”). Do check all.

English was corrected by a native speaker.